# Experimental Study on the *G*_max_ Characteristics of the Sand-Silt Mixed Soil Materials Using Bender Element Testing

**DOI:** 10.3390/ma15186200

**Published:** 2022-09-06

**Authors:** Jiang Bian, Hao Wu, Xing Xiao, Qi Wu, Zheng-Long Zhou

**Affiliations:** Institute of Geotechnical Engineering, Nanjing Tech University, Nanjing 211899, China

**Keywords:** sand-silt mixed soil materials, small-strain shear modulus, fines content, revised Hardin model

## Abstract

To study the small strain shear modulus (*G*_max_) of saturated sand-silt mixed soil materials, a series of tests were conducted using the bender element apparatus, and the influences of fines content (*FC*), relative density (*D*_r_), and effective confining pressure (${\sigma^{\prime }}_{3c}$) were taken into consideration. The test results indicate that the *G*_max_ of the mixed soil materials decreases first and then increases with the *FC* up to 100% with *D*_r_ = 35% and 50%, while the *G*_max_ decreases with the increasing *FC* when *D*_r_ = 60%. Moreover, for a given *D*_r_, the *G*_max_ increases with the increasing ${\sigma^{\prime }}_{3c}$, and the increase rate keeps constant under various *FC*s. The *G*_max_ of specimens under various *FC*s decreases with the increase of the void ratio (*e*). The decrease rate between the *G*_max_ and *e* differs when the ${\sigma^{\prime }}_{3c}$ is given, which is influenced by the *FC*. The *G*_max_ of the mixed soil materials can be evaluated by the Hardin model when the *FC* is determined. The best-fitting parameter *A* of the Hardin model first decreases and then increases as *FC* increases. The revised Hardin model, considering the influence of *FC*, ${\sigma^{\prime }}_{3c}$, and *e*, can be used to evaluate the *G*_max_ for different types of sand-silt mixed soil materials. The error between the evaluated and tested *G*_max_ is less than 10%.

## 1. Introduction

Small strain shear modulus *G*_max_, which is denoted as the shear modulus at small strain (i.e., below 10^−5^), is often used as an important mechanical index in soil material stability evaluation and numerical simulation [1,2]. Simultaneously, the angle of repose and packing fraction are most often considered as observables characterizing the mechanics of a granular material. The angle of repose is used for the determination of slope stability and design of retaining structures. Elekes and Parteli [3] established a model for predicting the angle of repose both on Earth and in other planetary environments by systematically considering the effect of sliding and rolling resistance, and nonbonded attractive particle–particle interactions. On the other hand, packing fraction can be used to develop a relationship between the particle-level interactions and the macroscopic structure [4]. In this paper, the dynamic properties of soil materials were investigated from the view of *G*_max_, regardless of the angle of repose and packing fraction.

*G*_max_ is influenced by various factors, such as void ratio (*e*) [5], particle properties [6,7,8], uniformity coefficient (*C*_u_) [9], fines content [10,11,12], fabric [13,14], stress history, and state [15,16], etc. Moreover, *G*_max_ is proved to be incredibly sensitive to *e* and effective confining pressure (${\sigma^{\prime }}_{3c}$). Considerable investigations have been performed on clean sand [6,17]. The fact is, most of the soil materials in nature are not pure sand or silt. Instead, sand-silt mixed soil materials with various fines content (*FC*, the mass percentage of soil with a particle size less than 0.075 mm) not only extensively exist in natural strata such as colluvium deposits, alluvial deposits, and glacial tills, but also have many applications in man-made constructions, including land reclamation, dams, and subgrades [18,19]. Therefore, the evaluation of sand-silt behavior as a common geomaterial-mixed material could be of particular interest. Many scholars have researched the impact of *FC* on *G*_max_. For instance, Iwasaki and Tatsuoka [20] first revealed that the *G*_max_ of Iruma Sand with the same *C*_u_ decreased with *FC* up to 15%. Yang and Liu [21] also observed a similar phenomenon in quartz sand with *FC* of up to 30%. Choo and Burns [22] performed bender element tests on sand-silt mixed soil materials and found that the shear velocity (*V*_s_) decreased with increasing *FC* under the same relative density (*D*_r_). Wichtmann et al. [23] conducted resonant column tests on quartz sand with different *C*_u_ and *FC* and established the empirical formulas for parameters *A*, *n*, and *c* with *FC* and *C*_u_. Cheng et al. [11] reported that the *G*_max_ of silty sand initially decreases and then increases with increasing plastic *FC* from 0 to 50%. Thevanayagam and Liang [24] and Goudarzy et al. [25,26] adopted the equivalent granular void ratio (*e*^∗^) instead of *e* in Equation (1). However, the relationships that many scholars have attempted to establish are discrepant and dependent on special sand and fines materials in specific areas. Therefore, the *G*_max_ behavior of mixed soil materials still deserves further investigation.

There exist various kinds of *G*_max_ test methods, such as the resonant column method and the bender element method. The bender element method is widely applied to the test *G*_max_, because of its simple principle, convenient operation, and non-destructive inspection [27,28]. A series of bender element tests were conducted to study the *G*_max_ of the saturated sand-silt mixed soil materials with a wider range of *FC* in this study than that in previous studies. The influences of *FC*, *D*_r_, and ${\sigma^{\prime }}_{3c}$ were taken into consideration. The empirical *G*_max_ evaluation model of various mixed soil materials was finally proposed.

## 2. Bender Element Test

### 2.1. Test Apparatus

The measurement of shear wave velocity (*V*_s_) and associated *G*_max_ was implemented using a pair of piezoceramic bender elements (BEs) installed in the GCTS HCA-300 (Tempe, AZ, USA) dynamic hollow cylinder-TSH testing system to excite the sample in the form of a sinusoidal wave. The test apparatus is shown in Figure 1. The confining and back pressure are measured by the standard pressure/volume controller. The axial static and dynamic forces can be controlled independently. The maximum range of the dynamic force can be 10 kN/5 Hz. The sensor for the axial force and displacement is placed at the top of the sample. The back pressure is applied at the top of the sample, and the excess pore water pressure is measured at the bottom of the sample. The testing principle of the bender element system is detailed in Hardin and Black [15] and Goudarzy et al. [26].

First, the *V*_s_ is calculated as following:(1)$$V_{s}=\frac{d}{t}.$$
where *d* is the effective distance of the shear wave propagation and its unit is m; *t* is the time of the shear wave propagation and its unit is s. The time domain method was used to determine *t* considering the simplicity and accuracy [29,30]. Then, the *G*_max_ value can be determined:(2)$$G_{\max}=\rho \cdot V_{s}{}^{2}$$
where the unit of *G*_max_ is MPa, the unit of *V*_s_ is m/s, and *ρ* is the dry density and its unit is kg/m^3^.

### 2.2. Test Material

The tested sand-silt mixed soil materials were taken from the tidal flat sediment in Nantong Gulf in China. The mixed materials with a particle size of less than 0.075 mm obtained by sieving are regarded as pure fines, and the rest of the particles are regarded as clean sand. The clean sand is a fine-grained, angular siliceous sand, and the pure fines is a non-plastic, sub-angular fines. Table 1 presents the basic index properties of the tested clean sand and pure fines according to ASTM D422, D4253, and D4254, and Figure 2 shows the particle size distributions of clean sand and pure fines. The non-plastic Nantong silt (pure fines) is added into the clean sands to achieve the mixed materials with *FC* from 0% to 100% by mass. The grain size distribution curves of the mixtures with various *FC* are also shown in Figure 2. The variation of the minimum void ratio (*e*_min_) and maximum void ratio (*e*_max_) versus *FC* of the mixtures is shown in Figure 3. It illustrates that when *FC* ≤ 30%, the values of *e*_min_ and *e*_max_ decrease with the increasing *FC,* and when *FC* > 30%, the values of *e*_min_ and *e*_max_ increase with the increasing *FC,* which indicates the presence of a threshold (around *FC* = 30%) for the effect of *FC* on *e*_min_ and *e*_max_.

### 2.3. Test Procedure

The tested specimen is a solid cylinder with a 100 mm diameter and a 200 mm height, and the moist tamping method was introduced to prepare the specimens [31]. All specimens were tested after saturation. Carbon dioxide flushing and de-aired water flushing were carried out first, then the back pressure saturation followed. When Skempton’s *B* > 0.95 [32], the specimen was considered fully saturated. Following saturation, each specimen was isotropically consolidated by keeping the effective principal stress ${\sigma^{\prime }}_{1c}$ = ${\sigma^{\prime }}_{2c}$ = ${\sigma^{\prime }}_{3c}$.

### 2.4. Testing Programe

In order to investigate the influences of *FC*, *D*_r_, and ${\sigma^{\prime }}_{3c}$ on *G*_max_ of the mixed materials. *FC* = 0, 10, 20, 30, 50, 70, and 100% were considered, and three specimens were prepared at different *D*_r_ (=35%, 50%, and 60%) at a fixed *FC*. The *G*_max_ was measured and subjected to ${\sigma^{\prime }}_{3c}$ at 100, 200, 250, 300, and 400 kPa in five stages, Table 2 details the test conditions and the corresponding *G*_max_. The typical signal wave of the bender element test of S11 specimen (${\sigma^{\prime }}_{3c}$ = 100 kPa) was illustrated in Figure 4. As shown in Figure 4, the *t* = 5.18 ms for the specific test condition can be determined by the time domain initial wave method clarity and precision.

## 3. Test Results and Analysis

### 3.1. Factors Influencing Maximum Shear Modulus

The measured *G*_max_ versus *FC* of the sand-silt mixed soil mixtures is illustrated in Figure 5. For the samples of 35% and 50% *D*_r_ under various ${\sigma^{\prime }}_{3c}$, the *G*_max_ decreased and then increased with the increasing *FC*, reaching the minimum when the *FC* was 30%. For the samples of 60% *D*_r_ under various ${\sigma^{\prime }}_{3c}$, the *G*_max_ decreased with the increasing *FC*. The *FC* of the inflection point of the *G*_max_ was changed with the increase of ${\sigma^{\prime }}_{3c}$. The reason can be that the transformation process of the contact state of the particles of the various *D*_r_ samples is different, which leads to the different interparticle structures [33]. When *FC* ≤ 30%, the fines will decrease the contact of skeleton particles formed by sand to form an unstable structure, which leads to a decrease in *G*_max_. When *FC* > 30%, the soil skeleton is borne by fine particles, and the shear resistance will recover.

The relationship between the *G*_max_ and ${\sigma^{\prime }}_{3c}$ of the samples is illustrated in Figure 6. For the samples under a given *D*_r_ and *FC*, the *G*_max_ almost increased linearly with the increasing ${\sigma^{\prime }}_{3c}$, which was in accordance with Salgado’s test results [34]. For the samples under various *D*_r_, the increase rate of the *G*_max_ with respect to ${\sigma^{\prime }}_{3c}$ was varied. Yang et al. [21] found that the increase rate of the *G*_max_ with respect to ${\sigma^{\prime }}_{3c}$ decreased when the samples were under a given *D*_r_ and *FC*. However, Wichtmann et al. [35] obtained a different conclusion. They found that the increase rate of the *G*_max_ with respect to ${\sigma^{\prime }}_{3c}$ increased when the samples were under a given *D*_r_ and *FC*. The opposite conclusions mean that the degree of association between the *G*_max_ and ${\sigma^{\prime }}_{3c}$ is related to the physical properties of the sand-silt mixed soil materials.

The relationship between the *G*_max_ and *e* of the samples were illustrated in Figure 7. For the samples under a given ${\sigma^{\prime }}_{3c}$ and *FC*, the *G*_max_ decreased with the increasing *e*. The decreasing rate of the *G*_max_ with respect to *e* was distinctly influenced by *FC* when the ${\sigma^{\prime }}_{3c}$ was the same. Yang et al. [21] found that the relationship between the *G*_max_ and *e* of the samples under various *FC* or ${\sigma^{\prime }}_{3c}$ was linear, and the decreased rate of the *G*_max_ with respect to *e* was unrelated to *FC*. The different conclusions mean that the correlation between the *G*_max_ and *e* is related to the material type of the mixed soils, which needs further study to find the influence factors.

### 3.2. The Evaluation Method of the G_max_ of the Sand-Silt Mixtures

The acknowledged empirical relationship among the ${\sigma^{\prime }}_{3c}$, *e*, and *G*_max_ of the non-cohesive soil is shown in Equation (3):(3)$$G_{\max}=AP_{a}F(e){\left(\frac{{\sigma^{\prime }}_{3c}}{P_{a}}\right)}^{n}$$
where, *A* is the parameter influenced by the property of the soil material type. *P*_a_ is the standard atmospheric pressure (≈100 kPa). *n* is the empirical constant, and the value is between 0.4 and 0.6 for the pluvial sand, and many researchers have observed that the value can be taken as 0.5 [35,36]. *F*(*e*) is the function of *e*, which is decreased with the increase of *e*. Hardin and Black 15 described the *F*(*e*) as Equation (4):(4)$$F\left(e\right)=\frac{{\left(c-e\right)}^{2}}{1+e}$$
where *c* is the parameter influenced by the shape of the soil particle, and the values are 2.17 and 2.97 for the angular and rounded sand particles, respectively. The shape of Nantong sand particles is angular, and the value of *c* is 2.17 as a result. The reasonability of Equation (4) is verified by many scholars, such as Iwasaki et al. [20], Yamashita et al. [37]. According to Equations (3) and (4), the Hardin evaluation model of the *G*_max_ of the Nantong sand-silt mixed soil materials can be described by Equation (5):(5)$$G_{\max}=AP_{a}\frac{{\left(c-e\right)}^{2}}{1+e}{\left(\frac{{\sigma^{\prime }}_{3c}}{P_{a}}\right)}^{0.5}$$

The normalized relationship between the *G*_max_~*F*(*e*)~${\sigma^{\prime }}_{3c}$/*P*_a_ is shown in Figure 8. The data of the *G*_max_~*F*(*e*)~${\sigma^{\prime }}_{3c}$/*P*_a_ is at the top when the *FC* is 0 (pure sand). The data of the *G*_max_~*F*(*e*)~${\sigma^{\prime }}_{3c}$/*P*_a_ is at the bottom when the *FC* is 30% (pure sand). The data of the *G*_max_~*F*(*e*)~${\sigma^{\prime }}_{3c}$/*P*_a_ is above the evaluation of the Hardin model when the *FC* is 100% (pure silt). The results mean that the influence of the *FC* on the relationship between *G*_max_~*F*(*e*)~${\sigma^{\prime }}_{3c}$/*P*_a_ is not monotonic. The fitted results of the *G*_max_~*F*(*e*)~${\sigma^{\prime }}_{3c}$/*P*_a_ data under various *FC* via the Hardin model are improper.

The relationship between the *G*_max_/*F*(*e*) and (${\sigma^{\prime }}_{3c}$/*P*_a_)^0.5^ of the sand-silt mixed soil materials under various *FC* is shown in Figure 9. The *G*_max_/*F*(*e*) increases almost linearly with the increased (${\sigma^{\prime }}_{3c}$/*P*_a_)^0.5^ when the *FC* is given. The *G*_max_ of the mixed soil materials under a given *FC* can be evaluated via the Hardin model. The data points (black square) and evaluation lines of the *G*_max_ of the mixed soil materials by the Hardin model are shown in Figure 9. It can be seen from the table that the *R*^2^ of mixture under various *FC* is almost all greater than 0.9, which means the *G*_max_ evaluation of the Hardin model is effective. In addition, the relationship between the parameters *A* and *FC* is not monotonous.

Thevanayagam and Martin [33] proposed the definition of the particle contact state of the mixed materials in order to describe the influence of *FC* on the mechanical behavior. The mixed materials are composed of sand and fines particles of various sizes, and the contact state of the sand and fines particles influences the mechanical behavior. The sand particles contact each other and constitute the skeleton of the mixed materials when the *FC* is low. The fines fill the voids left by the mixed materials. The mechanical behavior of the mixed materials is dependent on the framework of the sand particles, and the materials under such a particle contact state are called sand-like soil materials. The fines particles contact each other and constitute the skeleton of the mixed materials when the *FC* is high. The sand particles are suspended in the fines. The mechanical behavior of the mixed materials is dependent on the framework of the fines particles, and the mixed materials under such a particle contact state is called silt-like soil materials. As a result, there is a threshold fines content (*FC*_th_) for the mixed materials [33]. The mixtures are called sand-like soil materials when the *FC* is smaller than the *FC*_th_. The mixtures are called silt-like soil materials when the *FC* is larger than the *FC*_th_. The *FC*_th_ is a critical parameter which can distinguish the particle contact state and influence the mechanical behavior of the mixed materials. Rahman et al. [38] proposed an empirical function to determine the *FC*_th_.
(6)$$FC_{\mathrm{th}}=0.40\times \left(\frac{1}{1+\exp(\alpha -\beta \cdot \chi )}+\frac{1}{\chi }\right)$$

*α* and *β* are the fitted parameters and the values are suggested as 0.50 and 0.13, respectively. $\chi =d_{10}^{s}/d_{50}^{f}$, $d_{10}^{s}$ is the effective size of the sand and $d_{50}^{f}$ is the average size of the fines. The *FC*_th_ used in the study is about 35% which can be calculated according to the data of Table 1 and Equation (5).

The relationship between parameter *A* and *FC* is shown in Figure 10. Parameter *A* decreases with the increasing *FC* when the *FC* is smaller than *FC*_th_. Parameter *A* increases with the increasing *FC* when the *FC* is larger than *FC*_th_. As a result, parameter *A* can be described by Equation (6):(7)$$A\left(FC\right)=\left\{ \begin{array}{l} A_{\left(FC_{c}=0\right)}-mFCFC\leq FC_{\mathrm{th}}\\ A_{\left(FC=100\%\right)}-n\left(1-FC\right)FC>FC_{\mathrm{th}} \end{array}\right.$$

The *m* and *n* are the fitted parameters. *A*_(*FC*=0)_ is the parameter *A* fitted by Hardin model of pure sand. *A*_(*FC*=100%)_ is the parameter *A* fitted by the Hardin model of pure fines. The *m* and *n* are 1.850 and 0.799, respectively, for the mixed materials tested in the study.

As a result, the revised Hardin model, which considered the influence of *e*, ${\sigma^{\prime }}_{3c}$, and *FC*, can be described by Equation (7):(8)$$G_{\max}=A\left(FC\right)P_{a}F\left(e\right){\left({\sigma^{\prime }}_{3c}/P_{a}\right)}^{0.5}$$

The relationship between the tested *G*_max_ and the evaluated *G*_max_ by the revised Hardin model of the various mixed soil materials in the study is shown in Figure 11, and the error between them is less than 10%. In addition, the test data of Payan et al. [39] is used to verify the reasonability of the revised Hardin model. The test data of Payan et al. [39] is fitted by the revised Hardin model and the fitted parameters are shown in Table 3. The relationship between the tested *G*_max_ and evaluated *G*_max_ by the revised Hardin model of the data of Payan et al. is shown in Figure 12, and the error between them is less than 10%. As a result, the revised Hardin model can be used to evaluate the *G*_max_ of the mixed soil materials.

## 4. Conclusions

A series of bender element tests were conducted to study the *G*_max_ of the saturated sand-silt mixed soil materials, and the influence of *FC*, relative density *D*_r_ and ${\sigma^{\prime }}_{3c}$ were taken into consideration. The revised Hardin model was proposed to evaluate the *G*_max_ of the materials based on the test data. The *G*_max_ of the mixed materials increases linearly with the increased ${\sigma^{\prime }}_{3c}$ and the increase rate almost stays constant. The relationship between the *G*_max_ and *FC* is related to the *D*_r_ when the ${\sigma^{\prime }}_{3c}$ is given. The *G*_max_ decreases with the increased *e* and the decrease rate is influenced by the *FC* when the ${\sigma^{\prime }}_{3c}$ is given. The Hardin model can be used to evaluate the *G*_max_ of the mixed materials when the *FC* is given. The parameter *A* decreases and then increases with the increased *FC*. The revised Hardin model, which is based on the threshold fines content (*FC*_th_) can be used to evaluate the *G*_max_ of the mixed materials under various *FC*, *e,* and ${\sigma^{\prime }}_{3c}$, and the error between the evaluated and tested value is less than 10%. This study can provide reference data for seismic response analysis for Nantong sites with different levels of fine-grained and valuable references for other related research.

It should be noted that there may be differences in *G*_max_ between wet and dry soil materials. In engineering practice, the soil materials are mostly wet and especially saturated below the groundwater level. Thus, this study focuses on the *G*_max_ of wet soil materials, and the difference between wet and saturated soils will be investigated in the near future.

## Figures and Tables

**Figure 1 materials-15-06200-f001:**
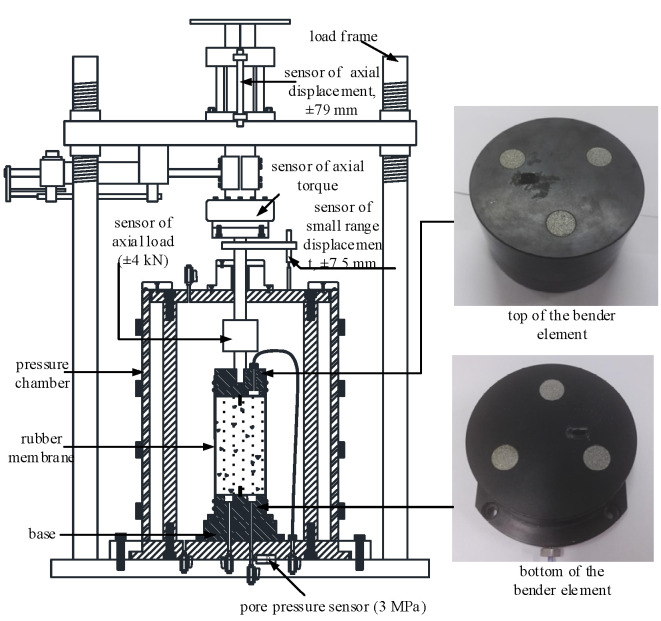
Bender element test apparatus.

**Figure 2 materials-15-06200-f002:**
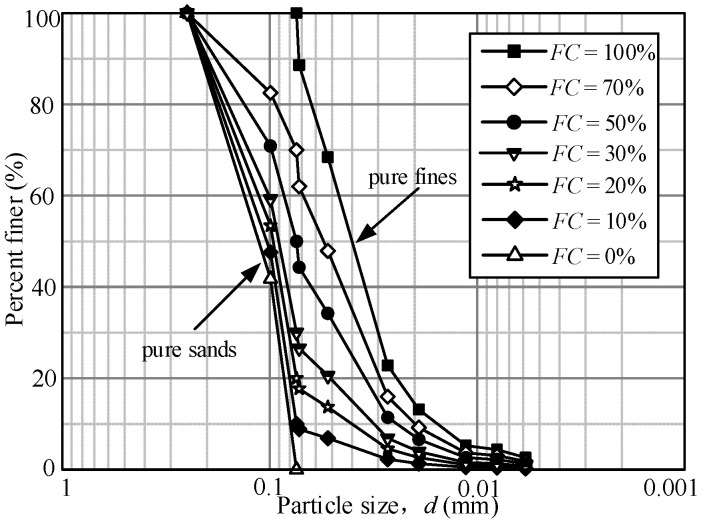
The particle size distribution curves of the sand-silt mixtures.

**Figure 3 materials-15-06200-f003:**
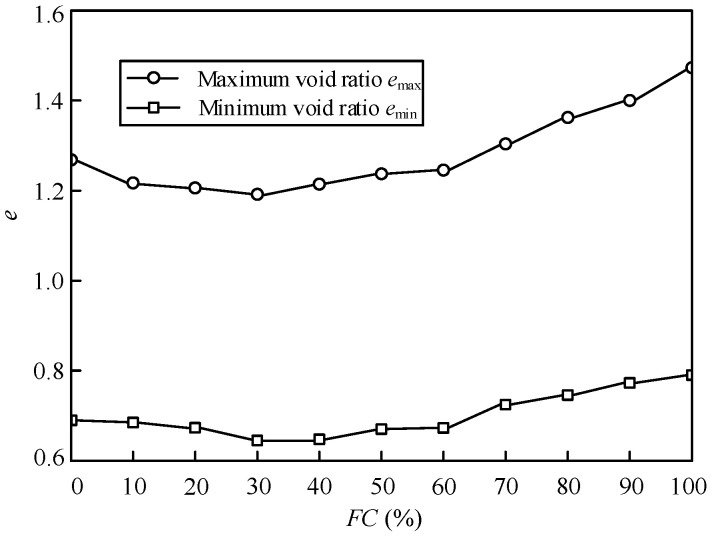
Variation of *e*_min_ and *e*_max_ versus *FC* of the sand-silt mixtures.

**Figure 4 materials-15-06200-f004:**
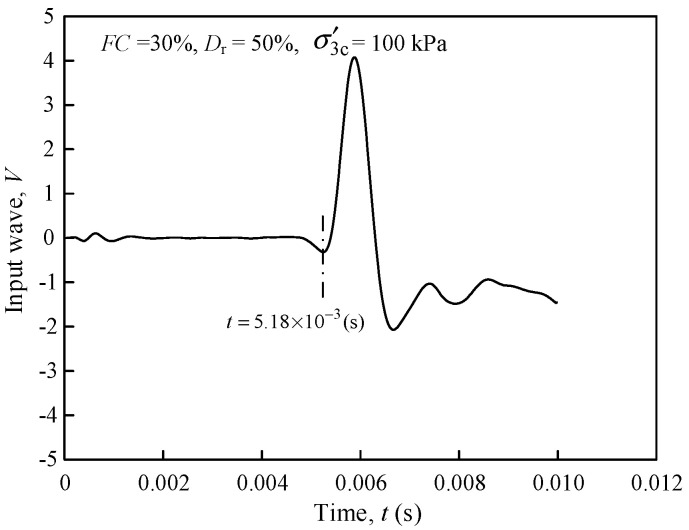
The typical signal wave of the bender element test.

**Figure 5 materials-15-06200-f005:**
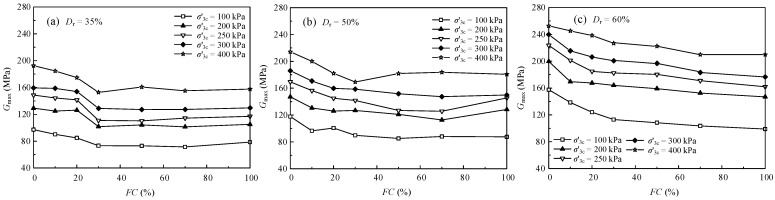
The relationship between the *G*_max_ and *FC.*

**Figure 6 materials-15-06200-f006:**
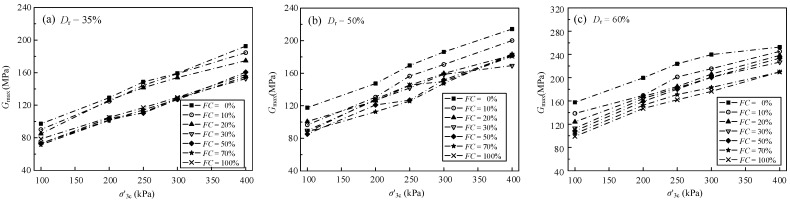
The relationship between the *G*_max_ and ${\sigma^{\prime }}_{}$.

**Figure 7 materials-15-06200-f007:**
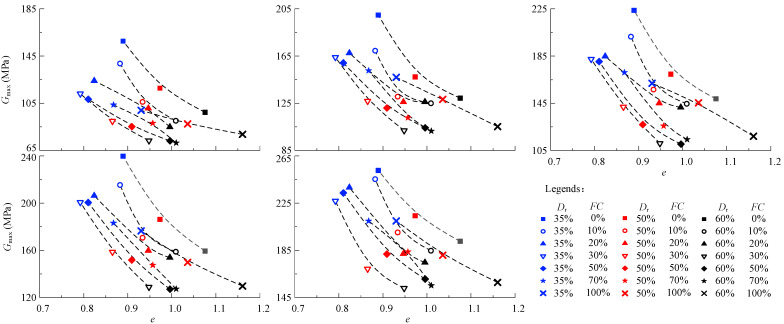
The relationship between the *G*_max_ and *e*.

**Figure 8 materials-15-06200-f008:**
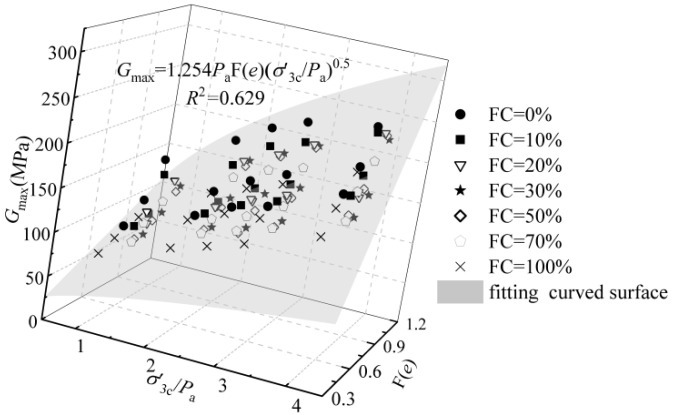
The relationship between the *G*_max_~*F*(*e*)~${\sigma^{\prime }}_{3c}$/*P*_a_ of the sand-silt mixed soil materials.

**Figure 9 materials-15-06200-f009:**
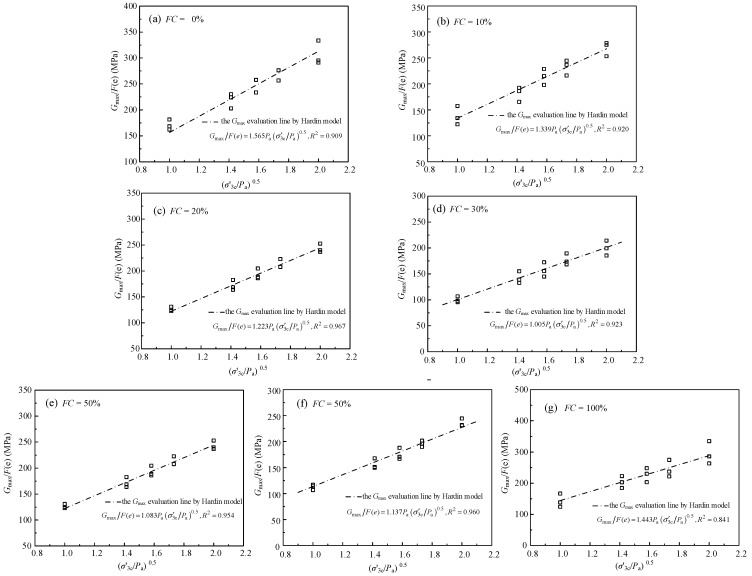
The relationship between the *G*_max_/*F*(*e*) and (${\sigma^{\prime }}_{3c}$/*P*_a_)^0.5^ the sand-silt mixed soil materials with different *FC.* The □ represents a data point.

**Figure 10 materials-15-06200-f010:**
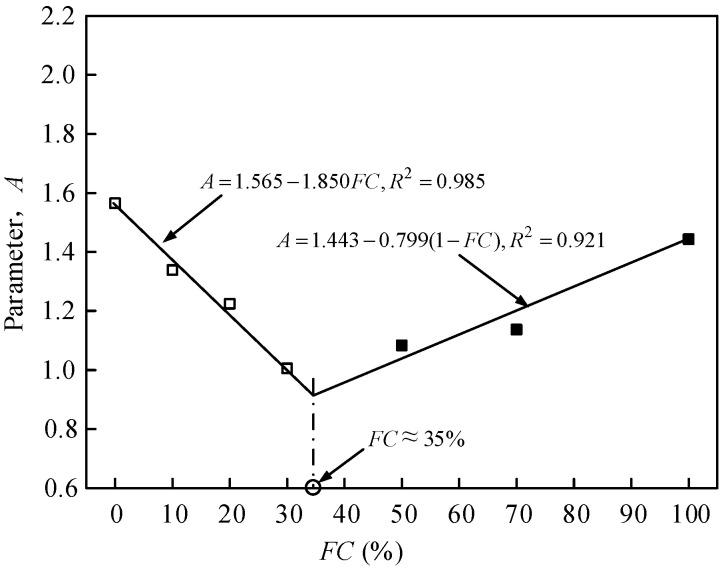
The relationship between the parameter *A* and FC of the sand-silt mixed soil materials. The □ represents a data point.

**Figure 11 materials-15-06200-f011:**
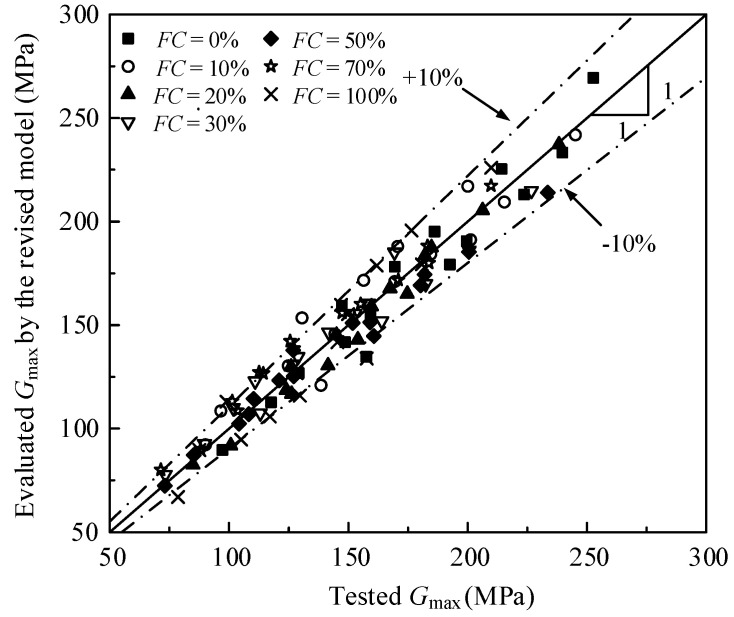
The relationship between the tested *G*_max_ and evaluated *G*_max_ by the revised model in the study.

**Figure 12 materials-15-06200-f012:**
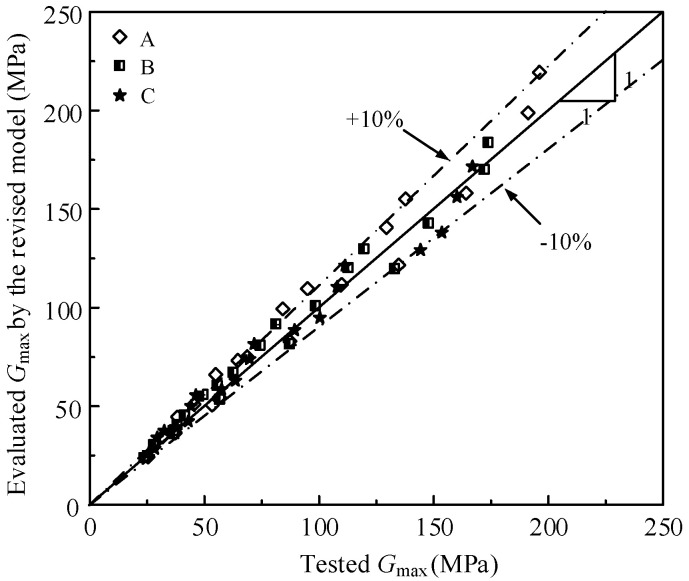
The relationship between the tested *G*_max_ and evaluated *G*_max_ by the revised model of Payan et al.

**Table 1 materials-15-06200-t001:** The basic physical properties of the sand and fines.

**Index**	*d*_50_/mm	*d*_10_/mm	*C* _u_	*G* _s_	*e* _max_	*e* _min_
**Sand**	0.114	0.080	1.672	2.672	1.262	0.662
**Fines**	0.040	0.016	2.931	2.719	1.481	0.764

Notes: *d*_50_—average particle size, *d*_10_—effective particle size, *C*_u_—nonuniform coefficient, *G*_s_—specific gravity.

**Table 2 materials-15-06200-t002:** The detailed test scheme of the bender element test.

No.	*FC*/%	*D*_r_ /%	*e*	*ρ* /(g/cm^3^)		*G*_max_/MPa				
					${\sigma^{\prime }}_{3c}/\mathrm{kPa}$	100	200	250	300	400
S1	0	35	1.076	1.286		97.2	129.2	148.7	159.3	192.6
S2	0	50	0.973	1.352		117.7	147.3	169.5	186.2	214.2
S3	0	60	0.890	1.412		157.6	199.7	223.7	239.7	252.7
S4	10	35	1.009	1.334		90.2	124.9	144.4	158.8	184.6
S5	10	50	0.934	1.386		96.6	130.6	156.5	170.7	200.1
S6	10	60	0.883	1.424		138.6	169.4	201.3	215.4	245.2
S7	20	35	0.936	1.348		84.9	126.2	141.5	154.0	174.7
S8	20	50	0.947	1.382		100.7	126.0	145.1	159.8	182.1
S9	20	60	0.824	1.475		124.0	167.5	184.8	206.3	238.3
S10	30	35	0.948	1.386		73.3	101.8	111.0	128.9	152.8
S11	30	50	0.865	1.448		89.8	126.8	141.8	158.6	169.3
S12	30	60	0.792	1.506		113.1	164.0	182.4	200.7	226.8
S13	50	35	0.996	1.358		73.1	104.2	110.3	127.1	160.7
S14	50	50	0.909	1.419		85.2	121.0	126.9	151.8	181.9
S15	50	60	0.810	1.497		108.3	159.1	180.3	200.5	233.6
S16	70	35	1.010	1.350		71.4	101.4	114.3	127.3	155.2
S17	70	50	0.957	1.387		88.1	112.7	125.7	147.4	183.8
S18	70	60	0.868	1.453		103.6	152.6	171.0	183.1	209.8
S19	100	35	1.231	1.258		78.7	105.1	117.1	129.7	157.7
S20	100	50	1.125	1.335		87.5	128.3	145.4	149.9	180.9
S21	100	60	0.990	1.409		98.9	146.9	161.9	176.5	209.8

**Table 3 materials-15-06200-t003:** The fitted parameters by the revised Hardin model of Payan et al.

Code	Test Material	FC_-th_	*A* _(FC=0)_	*A* _(FC=100%)_	*m*	*n*
A	White sand + quartz fines	33.3	0.675	0.385	1.170	0.150
B	Blue sand 1 + quartz fines	29.5	0.566	0.385	0.810	0.082
C	Blue sand 2 + quartz fines	42.7	0.541	0.385	0.957	0.442

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
