# Peer review of "Experimental Study on the Gmax Characteristics of the Sand-Silt Mixed Soil Materials Using Bender Element Testing"

_materials, 2022, doi:10.3390/ma15186200_

Round 1
Reviewer 1 Report
I think this paper is not ready for publication because the introduction should contain a paragraph telling about the research on granular materials (coarse and fine particles) more generally, including the focus given on the angle of repose and the packing fraction. Indeed, Gmax is one important quantity but the angle of repose and packing fraction are most often considered as observables characterizing the mechanics of a granular material: Edwards statistical mechanics for jammed granular matter.A Baule, F Morone, HJ Herrmann, HA Makse
Reviews of modern physics 90 (1), 015006 (2018)
https://doi.org/10.1103/RevModPhys.90.015006 An expression for the angle of repose of dry cohesive granular materials on Earth and in planetary environments F Elekes, EJR Parteli Proceedings of the National Academy of Sciences 118 (38), e2107965118 (2021) https://doi.org/10.1073/pnas.2107965118 I thus think that the authors should add this remark in the introduction, and cite the review paper about the theory of the packing fraction above (Baule et al., above) and an interesting model for the angle of repose of geomaterials, which appeared recently in PNAS (Elekes et al., above). The authors could even think whether they could do research to complement or improve the insights from the theoretical and experimental work considered above. This more generally introduction is very important, and the authors should add a brief comment to tell the reader what is learned about the granular material through Gmax that is not learned from the study of the angle of repose and the packing fraction. Section "Test apparatus", first sentence: What is wave velocity Vs? Section "Test procedure": "were isotropically consolidated": How? Please recall. Line 214: "The parameter A decreases with the increased A when the FC is smaller than FCth.". This sentence cannot be understood and should be rephrased/corrected. In the conclusions, it is not clear how this work is helping the scientific field of the paper. It looks like the main contribution is that you are measuring Gmax for a system including coarse and fine particles by covering a broad range of volume fraction ratios (coarse/fine). However, it is not clear whether or not such a study was made before. If yes, this should be acknowledged and the main difference between the present work and the previous study should be recalled. Moreover, you should emphasize in the conclusions possible differences between wet and dry materials. The paper cannot be accepted for publication now, but if the authors considered the comments above, then I think that the paper might become publishable after corresponding revision.
Reviewer 2 Report
materials-1881136
Title: Experimental study on Gmax characteristics of the sand-silt mixed soil materials using bender element testing
Authors: Bian Jiang , Wu Hao , Xiao Xing , Wu Qi , Zhou Zhenglong
Overview and general recommendation for the journal:
The article used the bender element (BE) test method to obtain the small strain shear modulus (Gmax) of sand-silt mixed soil materials. The authors also compared the measured Gmax with the Gmax simulated using existing models in the literature. I recommend a review of the paper. The following comments are made to the authors before the article is accepted for publication.
Title:
I would suggest the title "Experimental study on the Gmax characteristics of sand-silt mixed soil materials using bender element testing"
Introduction:
Equation 1: equations should be avoided in the Introduction. If the authors include all the equations used in the methodology, it will be easier for the reader. I recommend that only the new equations (proposed equations) be included in the results section.
Line 31: check font/size for where
Line 32: check parentheses without proper punctuation.
Line 38: "Considerable investigations have been performed on clean sand." Could you provide information (references) about these previous studies?
Line 44: common geomaterial would be better.
Line 49: insisted (maybe a better word?).
Bender element test:
Equation numbering needs to be revised.
Line 68: Variables must be declared at their first appearance in the text. For instance, see BE.
Line 80: Units of variables must be presented when the variables are declared to facilitate mathematical understanding. For instance, see d.
Figure 3: Could the authors explain the U-shape of the e versus FC curve?
Figure 4: Explain the peaks of V with time.
Test results and analysis:
Line 126: explain why Gmax is minimum when FC is 30\%.
Figure 7: should be stretched to improve readability.
Line 170: explain why the parameter n was set to 0.5.
Line 208/209: variable FCth is in a different format than the equation.
Line 214: The text between lines 214 and 216 is difficult to understand. Please, rephrase.
Conclusions:
How can it be possible to transfer the output of your research into engineering practice?
Language:
The text could be revised to improve readability. Here are some points that could be improved.
Line 51: established empirical formulas for.
Line 90: non-plastic.
Line 103: The tested specimen is a solid cylinder...
Line 113: and and
Line 133: The relationship is illustrated in Fig...
Line 136: was varied
Line 138: obtained a different conclusion
Round 2
Reviewer 1 Report
The authors revised their manuscript by taking my comments satisfactorily into account, and I can now recommend publication of this manuscript in Materials.
Reviewer 2 Report
As the authors modified the article accordingly, I have no further questions and recommend publication.